# How social evaluations shape trust in 45 types of scientists

**Vukašin Gligorić** *, **Gerben A. van Kleef, Bastiaan T. Rutjens**

Departement of Social Psychology, University of Amsterdam, Amsterdam, The Netherlands

* v.gligoric@uva.nl

## Abstract

Science can offer solutions to a wide range of societal problems. Key to capitalizing on such solutions is the public's trust and willingness to grant influence to scientists in shaping policy. However, previous research on determinants of trust is limited and does not factor in the diversity of scientific occupations. The present study ($N$ = 2,780; U.S. participants) investigated how four well-established dimensions of social evaluations (competence, assertiveness, morality, warmth) shape trust in 45 types of scientists (from agronomists to zoologists). Trust in most scientists was relatively high but varied considerably across occupations. Perceptions of morality and competence emerged as the most important antecedents of trust, in turn predicting the willingness to grant scientists influence in managing societal problems. Importantly, the contribution of morality (but not competence) varied across occupations: Morality was most strongly associated with trust in scientists who work on contentious and polarized issues (e.g., climatologists). Therefore, the diversity of scientific occupations must be taken into account to more precisely map trust, which is important for understanding when scientific solutions find their way to policy.

**Data Availability Statement:** The database and the R script used for analyses can be found at OSF: https://osf.io/d5zcj/.

**Funding:** This project has received funding from the European Research Council (ERC) under the European Union's Horizon 2020 research and

## Introduction

Scientists can offer valuable insights and possible solutions when faced with pressing environmental and societal problems such as climate change or the COVID-19 pandemic. However, trust in scientists is brittle. For example, during the COVID-19 pandemic trust in science and scientists has been documented to have decreased in several countries, including France, Italy, and the US [1, 2]. But also before the pandemic trust levels left much to be desired: In 2018, only 18% of the world population reported high levels of trust in scientists [3]. Given that trust in scientists is one of the key predictors of positive attitudes and compliance with science-based recommendations (e.g., vaccination [1] or pro-environmental behavior [4, 5]), insights into how trust in scientists is shaped and can be enhanced are crucial. However, the current understanding of trust in scientists is hampered by the implicit assumption that all scientific occupations are the same. This state of affairs hinders the successful implementation of science-based solutions to societal challenges. In the present research, we address this problem by investigating how trust is shaped by social evaluations across 45 scientific occupations, as well

innovation programme (grant agreement No. 849125 awarded to Bastiaan T. Rutjens). The funders had no role in study design, data collection and analysis, decision to publish, or preparation of the manuscript.

**Competing interests:** The authors have declared that no competing interests exist.

as how these social evaluations subsequently shape willingness to grant scientists influence in managing societal problems.

## Diversity of scientific occupations

Science is not a monolithic enterprise: It consists of a plethora of disciplines that each comes with its own goals, values, and approaches. It is conceivable, then, that perceptions of scientists and trust in scientists differ across scientific occupations. Indeed, previous research shows that the term "scientist" is an overgeneralization. In a systematic investigation of social evaluations of more than 30 scientific occupations in the US and the UK, Gligorić and colleagues (2022) found that people perceive meaningful differences between different scientific occupations [6]. For example, sociologists were regarded as less competent than neuroscientists, whereas zoologists and related occupations were seen as the most moral and sociable. These differences in social evaluations could be empirically captured in clusters (e.g., biomedical scientists, data scientists). Similarly, Altenmüller and colleagues (2024) showed that scientific occupations also differ in perceptions along the ideological line of liberalism-conservatism, so that some occupations (e.g., sociologists, climate scientists) are seen as more liberal than others (e.g., mathematicians, chemists) [7]. Trust, too, varies across different types of scientists: For example, people hold more positive perceptions of publicly (vs. privately) funded scientists [8], and some scientists work on issues that are caught in the crosshairs of heavily politicised public debate (e.g., climate science, vaccination), which can trigger science scepticism [9].

Crucially, how this diversity translates to trust in scientists is to date unknown. This is because most research investigating scientist perceptions has focused on the generic term "scientists". Some exceptions are disparate studies of one or two specific groups or categories, such as scientists working on genetic modification (GM) [10], chemists and biologists [11], and mathematicians [12]. To date, however, no systematic comparison of trust in different types of scientists has been conducted. This is problematic because it remains unknown whether interventions aimed at increasing trust should be tailored to specific (groups of) scientists or scientists in general.

## Trust in scientists

**Social evaluations as antecedents of trust.**   What are the factors that contribute to trust in scientists? Some researchers point to the importance of social evaluations. Notably, Fiske and Dupree (2014) suggested that even though scientists are seen as competent, their perceived lack of warmth might contribute to lower trust [13]. This work is based on the two-dimensional Stereotype Content Model (SCM; [14]), which posits two social evaluation dimensions–*competence* and *warmth*. Although SCM is one of the best-known social evaluation models, other models proposed different structures of social judgment, arguing for separation of morality and warmth (e.g., politicians can be perceived as friendly, but immoral; [15–17]), separation of competence (capability) and assertiveness (confidence) [15], or even adding a new dimension (conservative-progressive beliefs) [18]. In a recent adversarial collaboration, researchers who posited five different social evaluation models attempted to integrate these models [19–21]. Their integration suggested that further division of the Big Two factors (Agency/Vertical dimension and Communion/Horizontal dimension) into four facets (competence and assertiveness; morality and warmth) is the most appropriate structure of social judgment [19–21]. Such distinctions showed to be useful for understanding evaluations of scientists as well, given that research has shown that scientists are not only seen as very competent (intelligent, smart), but are also perceived as more moral than warm, as well as more competent than assertive [6, 22]. However, although social evaluation models are a promising path to

increasing understanding of trust in scientists [13], to our knowledge, no research has directly tested how social evaluations of groups subsequently relate to trust. In other words, work so far leaves the question of how well-established social evaluations of scientists shape levels of trust in them. Additionally, as mentioned before, it is unknown whether the role of social evaluations in trust varies across occupations.

Another, somewhat related, strand of research on trust (in scientists) comes from organizational psychology, which suggests that trust is composed of three components: ability (skill, expertise), benevolence (self-interest vs societal benefits), and integrity (fairness and honesty) [23]. Trust in scientists could be, in principle, somewhat different as laypeople are unable to understand different scientific information without specialized knowledge in that area. Yet, the three-component model of trust has also shown to be applicable within the context of scientists: Hendriks and colleagues (2015) found that scientists were evaluated on these three dimensions [24]. More recently, it has been suggested that the factor of "openness" should be added as a fourth component of trust in scientists [10]. However, due to high intercorrelations between components, neither study tested the relative contribution of these dimensions to trust. These intercorrelations are expected (and inevitable) given that they are seen as trust *components* (parts of trust), rather than factors influencing trust. With this in mind, the current work focused on social evaluations as predictors, because they 1) are more clearly conceptualized in the literature, 2) show lower intercorrelations, and 3) potentially *shape* trust, rather than merely representing *components* of trust. Importantly, we did not focus solely on trust ratings, but we also included the consequence of trust, which we operationalized using a novel influence granting task (see pilot studies).

## The present study

In the present study, we investigated trust in many different groups of scientists. Specifically, we tested whether and how four theoretical social evaluation dimensions (competence, assertiveness, morality, warmth; [19, 25]) contribute to trust across 45 different scientific occupations. Moreover, we examined how trust in turn shapes the willingness to grant scientists influence on the management of complex societal problems, by including a novel influence granting task (details about pilot studies conducted to develop this task can be found in the S1 File). Incorporating this task enabled us to investigate not only the influence of social evaluations on trust perceptions, but also to assess the downstream consequences of these perceptions in a more ecologically valid way.

## Method

### Participants

As pre-registered, we aimed to collect data from 2813 participants. This sample size was determined based on the notion that correlations stabilize at around 250 participants [26]. Since each participant rated four (out of 45) occupations (see procedure), we needed 2813 participants (= 250*45/4) so that each occupation would obtain ratings from approximately 250 participants. To account for anticipated exclusions (see below), we sampled 3246 participants from Prolific, selecting only US participants whose minimum approval rate was 95/100. Participation took around 10 minutes, and participants were paid £1.05 (approx. $1.16 or €1.20). As pre-registered, we excluded participants who failed an attention check question ($n = 334$), speeders (twice faster than the median of 610 seconds; $n = 82$), and multivariate outliers (Mahalanobis distance on each item of the dependent variable, $n = 50$), which left us with a final sample of $N = 2780$ (1333 men, 1382 women, 65 indicated "other"; $M_{age} = 39.03$, $SD_{age} = 14.93$). Regarding education, 0.7% indicated education less than high school, 25.3% had

completed high school, 12.7% were students, 42.3% had an undergraduate degree, and 19.1% had a graduate degree. The sample was slightly liberal ($M = 3.07$; $SD = 1.76$; range 1–7) and somewhat religious ($M = 3.03$; $SD = 2.09$; range 1–7). No analyses were performed before data exclusion.

### Procedure and materials

Ethical approval was obtained at the authors' university. We report all measures used in the study. After reading the information letter and signing the consent form, participants reported their demographics, political orientation, and religiosity. Next, each participant was asked to rate four scientific occupations on several attributes. The four occupations were randomly selected out of 45 occupations. The list of 45 scientific occupations was obtained from the previous study in which US participants were asked to generate as many scientific occupations as they could [6]. After rating one scientific occupation, participants moved to the next one, until they had rated four.

First, participants rated an occupation on four social evaluation dimensions which were presented on four separate pages (the order of dimensions was randomized). The four dimensions were competence, assertiveness, morality, and warmth [19]. Each dimension was measured with five items on a 7-point (from -3 to 3) bipolar scale [15, 27]. Examples of items for competence included pairs such as "incompetent–competent" and "unintelligent–intelligent", and items for assertiveness included pairs such as "has no leadership skills at all–has leadership skills" and "unconfident–confident". Morality was assessed with pairs such as "unjust–just" and "unfair–fair", while warmth included pairs like "cold–warm" and "uncaring–caring". Reliabilities for all four dimensions were high: Median Cronbach's alphas were $\alpha = .92$ (range .80 –.95), $\alpha = .87$ (range .83 –.91), $\alpha = .93$ (range .88 –.96), and $\alpha = .94$ (range .93 –.96) for competence, assertiveness, morality, and warmth respectively. All items measuring each dimension, as well as their intercorrelations, are given in the S1 Fig in S1 File.

After rating an occupation on the social evaluation scales, trust was measured using the following item: "How much do you trust [occupation]?" (1 = *do not trust at all* to 7 = *trust completely*) [1]. We opted for this approach to measure trust for several reasons. First, various large-scale research studies have used this operationalization [1, 3], which predicts a range of behaviors[28, 29] and allows participants to construct their own meaning of trust, rather than researchers imposing theirs. Second, a one-item measure has been shown to function well as a substitute for a multi-item measure of trust [30]. Third, our pilot studies, which included both one-item and a more specific four-item [31] measures of trust, indicate that these correlate strongly (median Pearson's $r = .74$ across different occupations and pilot studies). Finally, a single-item measure is cost-effective, which was of importance given the large number of participants and occupations in the study.

Next, participants completed the influence granting task (IGT)—a novel task that we designed to measure the willingness to grant scientists influence on managing societal problems (see the S1 File for pilot studies in which we pre-tested the task). We presented participants with the following scenario:

> Imagine there is a pressing problem in your country that is affecting every citizen. You have the complete power to make a decision about how to solve the problem. This problem is very complex and, therefore, to solve it, the help and advice of various types of scientists would be useful. If you were to make a final decision, how strongly would you value the input of the following parties? Note that points must sum up to 100.

Participants were then instructed to use sliders to distribute 100 points of decision power to different parties. The points had to total 100 (constant sum type of question), and participants distributed them to seven different parties which were shown in a randomized order: community leaders, politicians, citizens, friends, family, themselves, and the scientific occupation at hand. Given that we wanted to allow for comparisons of effects between different scientific occupations, we developed one scenario for all occupations. The rationale for this task was that granting influence can be understood as one of the key trusting behaviors, in that people are more willing to confer decision-making power to individuals or institutions they trust [23, 32–34]. On average (across all scientific occupations), participants distributed the points in the following descending order: scientists (25.5), citizens (18.1), community leaders (16.9), themselves (16.6), friends (8.0), politicians (7.5), family (7.3).

After rating four occupations, participants were presented with an attention check question which was similar to the IGT and was phrased in the following way: "Imagine there is a pressing problem in your country that is affecting every citizen. Note that this question is an attention-check question. Please select the option *myself* and move it [the slider] to the maximum." Next, participants were asked to report how attentive they were (1 = *not very attentive* to 5 = *extremely attentive*). Failing either of the attention checks warranted exclusion (for the second question, answering "not very attentive" was considered an attention check failure). Finally, we asked participants to indicate if they knew all occupations they were asked to rate (2465 reported they knew all occupations, 315 indicated they did not). The results remained unchanged when analyses were conducted with data from only participants who reported knowing all occupations ($n$ = 2465). Note that testing the robustness of results after excluding participants who did not know occupations was not pre-registered.

## Results

All reported analyses were pre-registered unless stated otherwise. Before answering our research questions on the relationship between social evaluations and trust, we first calculated ratings of competence, assertiveness, morality, warmth, trust, and IGT for each occupation. Ratings were estimated from a mixed model which included a random intercept for participants because each participant rated only 4 (out of 45) occupations. The table of ratings by occupation is given in Table 1. Overall, scientists evoked positive perceptions given that all social evaluations and trust ratings were above the mid-point of the scales (though note that political scientists and economists evoked noticeably lower ratings of morality and trust compared to other occupations).

As per a reviewer's suggestion, we compared the fit of two models of social evaluations (non-preregistered analyses): one with the four-factor (the one we currently use) and one with the two-factor (agency/communion) solution. Only the four-factor solution showed good fit (CFI = 0.946, TLI = .937, RMSEA = .077, SRMR = .052, AIC = 444328, BIC = 444811), but not the two-factor one (CFI = 0.748, TLI = .717, RMSEA = .163, SRMR = .140, AIC = 483346, BIC = 483792). Comparing the models showed that the four-factor solution was significantly better ($\chi 2\text{diff}(5)$ = 9592.5, $p < .001$).

### Social evaluations predicting trust and influence granting

Since participants rated different occupations (ratings were nested in participants and occupations), multilevel analyses were required. Before conducting analyses, we performed mean-centering within clusters (occupations), to disaggregate between and within effects, and focused only on the latter. We also standardized all variables to facilitate the interpretation of

**Table 1. Social evaluations, trust, and IGT ratings of scientific occupations in the study with standard errors in the brackets.**

| Occupation | Competence | Assertiveness | Morality | Warmth | Trust | IGT |
|---|---|---|---|---|---|---|
| Agronomist | 5.79 (0.04) | 5.08 (0.05) | 5.48 (0.05) | 5.04 (0.06) | 4.99 (0.06) | 23.89 (0.93) |
| Anthropologist | 5.96 (0.04) | 5.33 (0.05) | 5.54 (0.05) | 5.18 (0.06) | 5.08 (0.06) | 23.30 (0.95) |
| Archeologist | 5.96 (0.04) | 5.47 (0.05) | 5.48 (0.05) | 5.04 (0.06) | 5.21 (0.06) | 19.83 (0.93) |
| Astronomer | 6.14 (0.04) | 5.44 (0.05) | 5.54 (0.05) | 4.97 (0.06) | 5.36 (0.06) | 24.30 (0.95) |
| Astrophysicist | 6.39 (0.04) | 5.70 (0.05) | 5.53 (0.05) | 4.49 (0.06) | 5.41 (0.06) | 29.42 (0.96) |
| Biochemist | 6.30 (0.05) | 5.44 (0.05) | 5.47 (0.05) | 4.52 (0.06) | 5.19 (0.06) | 28.55 (0.97) |
| Biologist | 6.13 (0.04) | 5.40 (0.05) | 5.64 (0.05) | 5.13 (0.06) | 5.35 (0.06) | 27.98 (0.96) |
| Botanist | 5.90 (0.04) | 4.98 (0.05) | 5.69 (0.05) | 5.57 (0.06) | 5.23 (0.06) | 20.17 (0.93) |
| Chemist | 6.29 (0.04) | 5.54 (0.05) | 5.50 (0.05) | 4.50 (0.06) | 5.30 (0.06) | 28.39 (0.95) |
| Climatologist | 5.77 (0.04) | 5.40 (0.05) | 5.44 (0.05) | 5.15 (0.06) | 5.04 (0.06) | 28.29 (0.96) |
| Computer scientist | 6.26 (0.04) | 5.21 (0.05) | 5.24 (0.05) | 4.08 (0.06) | 5.05 (0.06) | 24.86 (0.95) |
| Data scientist | 6.13 (0.04) | 5.24 (0.05) | 5.34 (0.05) | 4.24 (0.06) | 5.02 (0.06) | 31.96 (0.95) |
| Ecologist | 5.94 (0.04) | 5.41 (0.05) | 5.70 (0.05) | 5.55 (0.06) | 5.27 (0.06) | 29.26 (0.94) |
| Economist | 5.70 (0.04) | 5.42 (0.05) | 4.68 (0.05) | 4.04 (0.06) | 4.28 (0.06) | 24.14 (0.95) |
| Entomologist | 5.88 (0.04) | 5.07 (0.05) | 5.45 (0.05) | 4.94 (0.06) | 5.04 (0.06) | 18.73 (0.93) |
| Environmental scientist | 5.84 (0.04) | 5.55 (0.05) | 5.71 (0.05) | 5.43 (0.06) | 5.28 (0.06) | 32.60 (0.94) |
| Epidemiologist | 6.12 (0.04) | 5.61 (0.05) | 5.59 (0.05) | 5.00 (0.06) | 5.34 (0.06) | 33.61 (0.95) |
| Food scientist | 5.75 (0.04) | 5.06 (0.05) | 5.11 (0.05) | 4.88 (0.06) | 4.74 (0.06) | 19.96 (0.94) |
| Geneticist | 6.09 (0.04) | 5.42 (0.05) | 5.35 (0.05) | 4.71 (0.06) | 5.07 (0.06) | 24.56 (0.96) |
| Geographer | 5.88 (0.04) | 5.11 (0.05) | 5.55 (0.05) | 4.93 (0.06) | 5.27 (0.06) | 20.71 (0.95) |
| Geologist | 6.00 (0.04) | 5.30 (0.05) | 5.61 (0.05) | 4.95 (0.06) | 5.25 (0.06) | 24.09 (0.95) |
| Hydrologist | 5.91 (0.04) | 5.19 (0.05) | 5.48 (0.05) | 4.93 (0.06) | 5.08 (0.06) | 22.16 (0.96) |
| Immunologist | 6.14 (0.04) | 5.49 (0.05) | 5.63 (0.05) | 5.04 (0.06) | 5.35 (0.06) | 30.77 (0.95) |
| Marine biologist | 6.21 (0.04) | 5.59 (0.05) | 5.88 (0.05) | 5.81 (0.06) | 5.54 (0.06) | 24.81 (0.94) |
| Mathematician | 6.33 (0.04) | 5.40 (0.05) | 5.48 (0.05) | 4.22 (0.06) | 5.27 (0.06) | 26.35 (0.96) |
| Medical researcher | 6.17 (0.04) | 5.64 (0.05) | 5.59 (0.05) | 5.01 (0.06) | 5.26 (0.06) | 34.54 (0.94) |
| Meteorologist | 5.66 (0.04) | 5.26 (0.05) | 5.33 (0.05) | 5.25 (0.06) | 4.85 (0.06) | 19.85 (0.93) |
| Microbiologist | 6.24 (0.04) | 5.39 (0.05) | 5.57 (0.05) | 4.67 (0.06) | 5.36 (0.06) | 28.55 (0.95) |
| Neuroscientist | 6.39 (0.04) | 5.90 (0.05) | 5.67 (0.05) | 4.72 (0.06) | 5.53 (0.06) | 29.88 (0.94) |
| Nuclear physicist | 6.44 (0.04) | 5.83 (0.05) | 5.39 (0.05) | 4.23 (0.06) | 5.22 (0.06) | 27.53 (0.96) |
| Nuclear scientist | 6.32 (0.04) | 5.69 (0.05) | 5.35 (0.05) | 4.25 (0.06) | 5.14 (0.06) | 27.06 (0.94) |
| Oceanographer | 6.07 (0.04) | 5.42 (0.05) | 5.72 (0.05) | 5.47 (0.06) | 5.39 (0.06) | 23.49 (0.96) |
| Paleontologist | 5.95 (0.04) | 5.31 (0.05) | 5.56 (0.05) | 5.02 (0.06) | 5.19 (0.06) | 20.08 (0.94) |
| Pharmacologist | 5.90 (0.04) | 5.27 (0.05) | 5.28 (0.05) | 4.72 (0.06) | 4.97 (0.06) | 21.88 (0.93) |
| Physicist | 6.34 (0.04) | 5.59 (0.05) | 5.45 (0.05) | 4.29 (0.06) | 5.33 (0.06) | 28.62 (0.95) |
| Physiologist | 5.92 (0.04) | 5.47 (0.05) | 5.49 (0.05) | 5.11 (0.06) | 5.13 (0.06) | 25.74 (0.96) |
| Political scientist | 5.26 (0.04) | 5.32 (0.05) | 4.30 (0.05) | 4.10 (0.06) | 3.71 (0.06) | 18.42 (0.93) |
| Psychologist | 5.76 (0.04) | 5.47 (0.05) | 5.56 (0.05) | 5.48 (0.06) | 4.89 (0.06) | 22.58 (0.93) |
| Rocket scientist | 6.51 (0.04) | 5.92 (0.05) | 5.51 (0.05) | 4.43 (0.06) | 5.38 (0.06) | 28.04 (0.93) |
| Seismologist | 6.02 (0.05) | 5.38 (0.05) | 5.58 (0.05) | 4.79 (0.06) | 5.23 (0.06) | 24.21 (0.98) |
| Sociologist | 5.54 (0.04) | 5.19 (0.05) | 5.40 (0.05) | 5.32 (0.06) | 4.67 (0.06) | 25.08 (0.94) |
| Statistician | 6.04 (0.04) | 5.31 (0.05) | 5.38 (0.05) | 4.10 (0.06) | 5.01 (0.06) | 26.72 (0.95) |
| Virologist | 6.09 (0.04) | 5.46 (0.05) | 5.50 (0.05) | 4.73 (0.06) | 5.13 (0.06) | 31.36 (0.95) |
| Volcanologist | 6.01 (0.04) | 5.43 (0.05) | 5.41 (0.05) | 4.86 (0.06) | 5.24 (0.06) | 23.07 (0.95) |
| Zoologist | 6.00 (0.04) | 5.48 (0.05) | 5.93 (0.05) | 6.09 (0.06) | 5.48 (0.06) | 21.68 (0.93) |

*Note.* Means were estimated from a mixed model with a random intercept for participants.

**Table 2. Correlations between scientists' evaluation variables, trust, and influence granting (IGT).**

| | 1. | 2. | 3. | 4. | 5. |
|---|---|---|---|---|---|
| 1. Competence | | | | | |
| 2. Assertiveness | .651 | | | | |
| 3. Morality | .672 | .622 | | | |
| 4. Warmth | .423 | .482 | .637 | | |
| 5. Trust | .525 | .442 | .640 | .434 | |
| 6. IGT | .226 | .218 | .236 | .141 | .328 |

*Note.* All correlations are significant at the $p < .001$ level. Due to data transformation, degrees of freedom for correlations are $df$ = 11118 (2780 participants x 4 occupations = 11120)

the coefficients and enable comparing effects in predicting trust and influence granting. The correlation between the variables is given in Table 2.

We ran a multilevel model to investigate how social evaluations relate to trust. Because trust levels could be different for different participants and different occupations, we first ran a model with a random intercept for participants and occupations, and fixed effects for social evaluation measures. The results are given in Table 3 (left). There are several things to note. First, morality seems to play the most important role in shaping trust perceptions, followed by competence. Assertiveness and warmth also contributed, but to a smaller extent. Regarding random effects, random intercepts suggest that trust levels varied across participants and across occupations. The full random intercept model (AIC = 21329) showed better fit than the models without a random intercept for either participants or occupations (AIC = 24065, LRT (1) = 2738.3, $p < .001$ and AIC = 23312, LRT(1) = 1984.8, $p < .001$ respectively), indicating that trust levels were indeed different across participants and occupations.

As a next step, we aimed to test whether the effect of social evaluation dimensions (competence, assertiveness, morality, and warmth) predicting trust varied by occupation. We did so by creating a model that allowed slopes for each of the social evaluation dimensions to vary across occupations. Fixed effects remained relatively unchanged, with variances of slopes being .001 for competence, .000 for assertiveness, .009 for morality, and .003 for warmth.

**Table 3. Multilevel model (with random intercept for participants and occupations) in which social evaluation measures predict trust and IGT (influence granting) scores.**

| | Trust | | IGT | |
|---|---|---|---|---|
| | Fixed effects | | | |
| | $\beta$ (Standard error) | $t$ value | $\beta$ (Standard error) | $t$ value |
| Competence | .19 (.01) | 20.53*** | .10 (.01) | 10.72*** |
| Assertiveness | .05 (.01) | 5.54*** | .08(.01) | 8.99*** |
| Morality | .44 (.01) | 45.34*** | .14 (.01) | 13.88*** |
| Warmth | .08 (.01) | 9.73*** | .01 (.01) | 1.64 |
| | Random effects | | | |
| $\tau_{00 \text{ participant}}$ | .24 | | .63 | |
| $\tau_{00 \text{ occupation}}$ | .07 | | .03 | |
| ICC | .53 | | .72 | |
| Marginal $R^2$ / | .44/ | | .09/ | |
| Conditional $R^2$ | .74 | | .74 | |

*Note.* *** $p < .001$, $\tau_{00}$ = intercept variance, ICC = Intraclass correlation

Comparing models with and without random effects for each social evaluations dimension showed that the effects of competence and assertiveness on trust did not vary across occupations (LRTs(5) < 9.02, ps > .10), while the effect of morality (LRT(5) = 69.43, p < .001) and warmth (LRT(5) = 20.85, p < .001) did. A visual representation of the effects of social dimensions predicting trust across occupations is provided in Fig 1(A) and 1(B). In short, results suggest that the effects of competence and assertiveness predicting trust did not vary across scientific occupations, whereas the effects of morality and warmth did.

Next, we conducted the same analysis with the IGT as the dependent variable. Again, morality and competence were found to be more important than warmth and assertiveness, although the difference between their respective importance is noticeably smaller compared to the effects predicting trust (Table 3, right). As with trust, model comparisons indicated that IGT levels varied across participants and occupations: A full random intercept model (AIC = 21330) showed a better fit than models without either a random intercept for participants or occupations (AIC = 30367, LRT(1) = 7049.1, p < .001 and AIC = 24182, LRT(1) = 864.6, p < .001 respectively). To investigate the random effects of social evaluation dimensions predicting IGT, we applied the same strategy as for trust. Again, fixed coefficients remained the same, with variances of slopes being .002 for competence, .002 for assertiveness, .002 for morality, and .002 for warmth. Comparing models with and without random effects for each social evaluations dimension indicated that the effects of competence (LRT(5) = 12.8, p = .03), assertiveness (LRT(5) = 17.0, p < .01), and warmth on IGT (LRT(5) = 12.5, p = .03), but not morality (LRT(5) = 10.0, p = .07) varied across occupations. Given the low variances and marginal p values, these results did not decisively show whether including random effects is

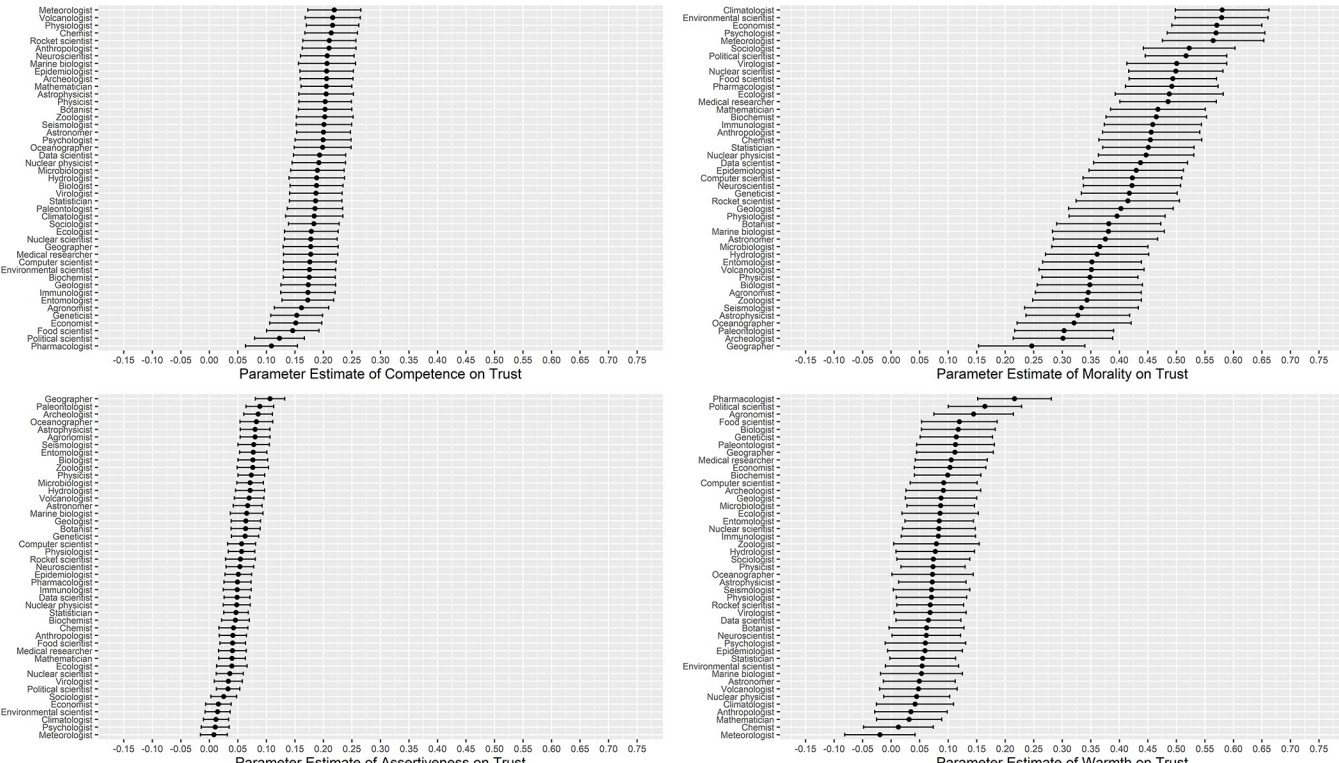

**Fig 1. A.** The estimates (beta coefficients, with 95% confidence intervals) of competence and assertiveness predicting trust for all 45 scientific occupations. The estimates are uniform across different occupations. **B.** The estimates (beta coefficients, with 95% confidence intervals) of morality and warmth predicting trust for all 45 scientific occupations. The estimates vary across different occupations.

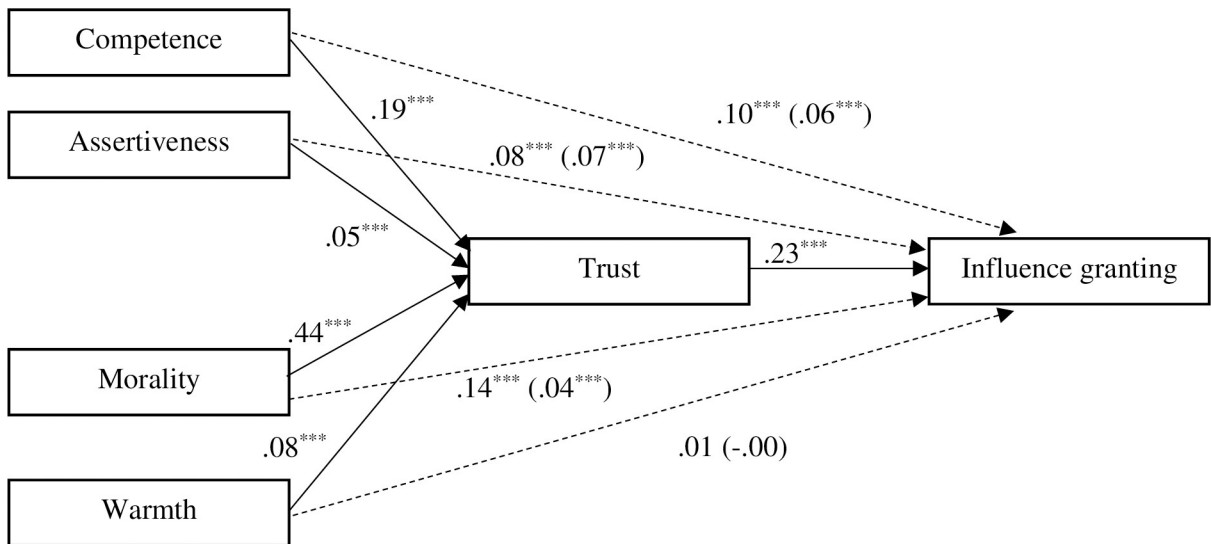

**Fig 2. The effect of social evaluations predicting influence granting via trust.** The model is averaged across occupations and participants with random intercepts for both factors. Trust partially mediated the effects of competence, assertiveness and morality. Significant paths are marked with three asterisks (***$p < .001$). Direct effects are given within brackets.

warranted. We, therefore, investigated the AIC of the models when each of the random slopes was dropped. These models had either equal or higher AIC than the baseline model with no random slopes (AIC = 23271). For this reason, we opted for selecting the simpler model, i.e., without random effects. A visual representation of the effects of social evaluations predicting IGT across occupations, which is provided in the S2 and S3 Figs in S1 File, also supported this decision. In short, these analyses show that the impact of social evaluation dimensions (competence, assertiveness, morality, warmth) on influence granting did not depend on occupation, unlike what was observed for trust ratings. This suggests that, for example, perceptions of morality have the same impact on willingness to grant scientists influence, regardless of the scientific occupation in question. Note that all analyses above retained the same pattern when the morality dimension was calculated excluding the "trustworthy" adjective due to semantic overlap (fully reported in the S1 File).

## Trust as a mediator of the effect of social evaluations on influence granting

Finally, we tested if trust mediates the effect of social evaluations predicting influence granting. To do so, we conducted a multilevel mediation analysis. This analysis allowed us to test whether trust mediates the relationship on average (ignoring different occupations), and how this indirect effect varies across different occupations. The coefficients were estimated from two regression models (first: social evaluations predicting trust; second: social evaluations and trust predicting IGT) which included a random intercept for participants and occupations and fixed effects of social evaluations. The significance of indirect effects (all significant, $p < .001$) was tested using bootstrapping (estimating the model 1000 times). Effects averaged across occupations are given in Fig 2. As evident from the figure, the effects of competence, assertiveness, and morality were partially mediated by trust, while warmth did not have a total effect in the first place. Mediation proportions (the ratio of the total effect that is accounted by the mediating variable) of competence, assertiveness, and morality effects were 42%, 14%, and 73% respectively. Random effects by occupation can be found in the S1 File.

We ran all the analyses above (non-preregistered) using only the big two factors–agency (calculated as an average of competence and assertiveness) and communion (calculated as an average of warmth and morality). This showed that both big factors contributed to predicting trust which mediated the effect on influence granting (S4 and S5 Figs in S1 File). Importantly, comparing the results of analyses with two vs. four factors allowed us to discern which parts of the agency and communion contribute to trust (competence and morality) and which do not (assertiveness and warmth).

## Discussion

Research on trust in scientists—arguably a key prerequisite for public acceptance of science-based solutions—is limited because it routinely treats scientific occupations as homogenous. The present study is, to our knowledge, the first to systematically investigate the antecedents of trust in scientists across a wide range of science domains. More specifically, we investigated how social evaluations shape trust as well as the willingness to grant scientists influence in managing societal problems. We found that trust in scientists was relatively high, with all occupations scoring above the mid-point. However, some scientists were trusted more than others. We discovered that perceptions of competence and morality play a prominent role in shaping trust ratings, more so than perceptions of assertiveness and warmth. Importantly, whereas the effect of competence was uniform, the effect of morality varied across occupations–morality was more strongly associated with trust in occupations working on contentious, publicly debated topics (see below).

The prominent role of competence and morality in trust suggests that trust in scientists has two requirements: Although a scientist must be knowledgeable and competent, this competence must be paired with good intentions (i.e., the scientist needs to be perceived as moral). Both ingredients are necessary to build a trustworthy scientist: Scientists would not be seen as scientists if they were perceived as incompetent [6, 13, 22], while morality is arguably the most important predictor of trust [10, 24]. Whereas Fiske and Dupre (2014) suggested that the perception of warmth (in terms of the two-dimensional SCM) is important for trust in scientists [13], our study advances this notion by showing that this particularly applies to morality, and not warmth. It is also noteworthy that competence (vs. assertiveness) and morality (vs. warmth) as strongest predictors are more stable and central to one's character as they refer to perception how people are (vs. how they behave/seem) [27].

Trust in scientists is likely different from trust in other social and occupational categories. A key defining characteristic of scientists is competence [6] and much of the work that scientists do is not fully comprehensible to laypeople, who therefore will have to rely on scientists' competence *and* their good intentions. However, trust in other groups might show different patterns. Indeed, research conducted within the SCM framework shows that trust in negotiators [35] and strangers [36] is determined by warmth, but not competence. Similarly, perceived morality (vs competence) is more important in cases such as group acceptance and rejection [37], and group evaluations [17]. Overall, this suggests that competence will contribute to trust when it is particularly relevant for an outcome, as is the case with scientists (e.g., discovering a drug to combat cancer depends on medical researchers' skills and knowledge).

Importantly, our study provides evidence that not all social evaluations shape trust in the same way for different occupations. Whereas competence similarly shaped trust regardless of scientific occupation, morality did not. Why would perceptions of morality have a weaker effect on trust for some occupations, but a stronger effect for others? This might be because certain groups of scientists work on more contentious and polarized issues. For instance, some of the occupations for which morality most strongly influenced trust were those in climate

science and politico-economic research. This finding aligns with the idea that many science attitudes are rooted in ideologies, identities, and other motivational factors [38, 39]. Additionally, some branches of science involve larger moral implications, as evidenced by public discussions about various scientific topics. This certainly applies to climate science [40, 41], nuclear physics [42], political science, and—especially since 2020—virology [43]. Whenever science is seen as especially relevant to people's lives, the perceived morality of the scientists involved arguably matters more, which is what the current data seem to support.

It is, however, important to note two differences when using social evaluations to predict trust perceptions, as compared to predicting the Influence Granting Task. First, the effect sizes were stronger when predicting trust perceptions. Second, for trust perceptions (but not IGT), the effect of morality perceptions varied across scientific occupations. We believe these differences emerged because social evaluations more closely matched trust perceptions in that both can be seen as attitudinal measures, whereas the IGT is a more behavioral measure (behavioral willingness). Another likely reason is common-method variance: seven-point scales were used to measure social evaluations and trust perceptions, but not for responses on the IGT. Finally, in responding to the IGT, factors other than trust might have influenced the responses (e.g., certain personality traits could drive participants more towards the option "myself"; individuals that have stronger ties with family could prefer the family option, etc.).

## Limitations, future research, and conclusion

Our study is not without limitations. Firstly, our measure of trust perception is short (one item) and might have seemed somewhat general to participants. Nevertheless, this approach to measuring trust is relatively common in research on trust in scientists [1, 3] as it provides a cost-effective way to capture key perceptions. Future research could put more focus on what trust in scientists entails, as well as on finding valid ways to investigate its consequences. Similarly, the utilized influence-granting measure may have come across as somewhat artificial, as the problem was unnamed, and participants were offered to grant influence to only one type of scientist. Therefore, participants might have had more specific problems in mind (e.g., climate change, COVID-19) when responding. However, it would be very difficult to come up with particular scenarios appropriate for each scientific occupation, especially given potential confounds. By keeping the phrasing of the question identical, we avoided this problem and were able to make comparisons across different occupations. Future research might benefit from utilizing a measure gauging the consequences of trust that is further improved on ecological validity, for example by assessing real-world behavior in specific domains. Another limitation of the study is that it utilized US participants, so the generalizability of the roles of competence and morality remains to be tested. Testing whether these results apply to other countries could be one avenue for future research. Finally, given the correlational nature of our study, a fruitful next step would be to experimentally assess the (relative) impact of competence and morality on trust and its consequences across scientific occupations, which would also pave the way for potential interventions (e.g., emphasizing scientists' good intentions in science communication).

In conclusion, the current work shows that trust in scientists varies considerably across occupations. Across 45 scientific occupations, trust was largely based on how competent and moral people perceived a scientist to be. The importance of morality was, however, not uniform across domains. For example, morality was not as important for trust in geographers as it was for trust in pharmacologists. These findings demonstrate that it is important to consider the diversity of scientific occupations when investigating trust and its contributing factors and

the willingness to grant scientists influence in engaging with societal problems and policymaking.

## Supporting information

**S1 File. Pilot studies.**
(DOCX)

## Author Contributions

**Conceptualization:** Vukašin Gligorić, Gerben A. van Kleef, Bastiaan T. Rutjens.

**Data curation:** Vukašin Gligorić.

**Formal analysis:** Vukašin Gligorić.

**Funding acquisition:** Bastiaan T. Rutjens.

**Investigation:** Vukašin Gligorić, Gerben A. van Kleef, Bastiaan T. Rutjens.

**Methodology:** Vukašin Gligorić, Gerben A. van Kleef, Bastiaan T. Rutjens.

**Project administration:** Vukašin Gligorić.

**Software:** Vukašin Gligorić.

**Supervision:** Gerben A. van Kleef, Bastiaan T. Rutjens.

**Visualization:** Vukašin Gligorić.

**Writing – original draft:** Vukašin Gligorić.

**Writing – review & editing:** Vukašin Gligorić, Gerben A. van Kleef, Bastiaan T. Rutjens.

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
