## [Decision Letter · Decision Letter 0]

17 Oct 2023

PONE-D-23-21811How Social Evaluations Shape Trust in 45 Types of ScientistsPLOS ONE

Dear Dr. Gligorić,

Thank you for submitting your manuscript to PLOS ONE. After careful consideration, we feel that it has merit but does not fully meet PLOS ONE’s publication criteria as it currently stands. Therefore, we invite you to submit a revised version of the manuscript that addresses the points raised during the review process.

We look forward to receiving your revised manuscript.

Kind regards,

Claudia Noemi González Brambila, Ph.D.

Academic Editor

PLOS ONE

Journal Requirements:

Reviewers' comments:

Reviewer's Responses to Questions

**Comments to the Author**

1. Is the manuscript technically sound, and do the data support the conclusions?

Reviewer #1: Partly

Reviewer #2: Yes

Reviewer #3: Yes

2. Has the statistical analysis been performed appropriately and rigorously? 

Reviewer #1: Yes

Reviewer #2: Yes

Reviewer #3: N/A

3. Have the authors made all data underlying the findings in their manuscript fully available?

Reviewer #1: Yes

Reviewer #2: Yes

Reviewer #3: Yes

4. Is the manuscript presented in an intelligible fashion and written in standard English?

Reviewer #1: Yes

Reviewer #2: Yes

Reviewer #3: Yes

5. Review Comments to the Author

Reviewer #1: I appreciate the scope of the manuscript and the underlying idea but I also wish the authors had made a number of different decisions.

1. I can't understand why the authors use the Abele and Wacjcizke approach to measuring people perceptions but then insist on talking about trust. In this regard, though I don't fully understand why, Abele and his colleagues seem to avoid using the word trust to talk about their research but they seem to only use the word 'trust' in passing (and seem to equate 'trustworthy' with integrity, which almost certainly creates measurement confusion).

And, in using Abele et al, I also wish you'd reported a CFA to show that the two facets for each of the big two are, indeed, distinct (noting that Abele does not report a CFA in the article cited, either). I have a lot of doubts about the degree to which the two facets are consistently distinct, though it's interesting that one facet in each dimension is about how someone IS (i.e., moral, or able) whereas the other dimension seems to be about how they SEEM (i.e., warm, assertive).

For the current purposes, I'd prefer to see a CFA in an appendix or a full measurement model as the primary results. I think it might also be worth reporting a version of the results that just uses the big two. You might be missing something by measuring competence and assertiveness separately, for eample.

2. I can't understand why you measured your criterion variable of 'trust' with such a weak measure. In this regard, you might note that Fiske's SCM has a nice set of outcomes that could have been used. Even better, however, would have been to rely on more-standard-than-DPM Integrative Model of Organizational Trust (which you cite) which makes a really nice distinction between trustworthiness perceptions (i.e., appraisals or beliefs) and trust BEHAVIORS as making oneself vulnerable. From this perspective, the trustworthiness perceptions exist in someone's head whereas behavioral trust exists in the world. In contrast, just asking someone 'how much do you trust someone' isn't clearly a perception or a behavior; it's just too vague a question and is thus going to capture a whole bunch of variance while providing little real insight.

With regard to to your IGT, you seem to be trying to recreate the wheel a bit, with all your work to make a criterion variable that you could have adapted from existing research behavioral trust discussions and avoided the use of a single-item measure.

Ultimately, my suggestion is to redo everything and ditch the "how much do you trust" question entirely. It just adds noise. You should also recognize that a limitation of your work is that you should have put more attention into measuring the criterion variable.

3. With regard to the criterion variable itself, I wish you had made it a little more specific. You say you tried to write it so that it fit every different type of scientist but, in doing so, you also made it a bit weird in that it's not clear from what's written whether the scientist is providing advice on a topic that's relevant to their area of expertise. The trust/trustworthiness literature in this regard makes the point that the expertise, for example, needs to match the behavior. The fact that someone knows a lot about meteorology isn't especially relevant to whether I would trust them to provide advice on vaccine saftey. The problem you describe, in this regard, seems to something like climate change (vs. a vaccine, or banking regulations).

Also with regard to the criterion variable, I would really like to have seen your initial work look at other potential correlates that specifically relate to behaviors of trust. With regards to what's reported, it'd be nice to show how other groups generally scored, though I'm not sure if there's a parsimonious way to do that.

More minor things

- Better label the figures (e.g., the X axis could be Parameter Estimate of Competence on IGT). If you don't read the note first then it'd be easy to think that the figures are means. And, it'd be really nice to have the means in the main body. Given PLOS is online, I'd also suggest making each of the figures bigger.

- Maybe tone down the use of the word 'effects' and use things like relationships or correlation or estimates to recognize that you're not doing an experiment

-I do not understand table s2

- table s3 could have SEs

-Give all your measures, not just examples; provide a correlation matrix of all of 16 in the supplementary files, if they're really distinct.

Reviewer #2: The manuscript addresses a relevant question, namely, the psychological determinants of trust in scientists. The (pre-registered) approach is transparent as well as comprehensible. The measurement of trust (on one item) could have been more differentiated, e.g. by measuring trust determinants as introduced by Mayer et al. 1995. For me, it remains unanswered why social evaluation has such a different effect on self-reported trust vs. behavior-based trust (IGT). This could have been discussed in more detail in the discussion. Overall, a highly relevant as well as methodologically clear contribution. I recommend to accept the submission.

Reviewer #3: Thanks for the opportunity to read firsthand about this research. Overall, the research topic is interesting. I have the following suggestions for the author’s reference:

1. The study design should be clearer, why is each person assessed against four randomly selected occupations? And where should each direct impression subject come from? It is suggested that the author should clarify the content.

2. After all, the subjects who answer the questions are very different. It is recommended that the authors conduct a differential analysis on the attributes of the people who answer the questions. In addition, it is recommended that the narrative description of the attributes of the respondent be presented in a table, which will be more clear.

3. The results should be discussed more fully.

4. The review of the literature should make a stronger case for why it is appropriate to select these traits for scientists. It must also be explained why it matters whether scientists trust non-scientists.

6. PLOS authors have the option to publish the peer review history of their article (what does this mean?). If published, this will include your full peer review and any attached files.

Reviewer #1: No

Reviewer #2: No

Reviewer #3: No

---

## [Author Response · Author response to Decision Letter 0]

18 Dec 2023

Dear Dr. Claudia Noemi González Brambila,

We would like to thank you and the reviewers for the valuable feedback and for inviting us to submit a revision. We have done our best to address the points the reviewers raised, and we revised the manuscript accordingly. Below, we respond to each point and highlight how we changed the manuscript. For readability purposes, our responses in the letter are provided in blue, while the changes in the manuscript are highlighted in yellow. 

Thank you again for the opportunity to revise our manuscript. We hope that this revision addresses the comments raised sufficiently. We look forward to hearing from you again. 

With best regards,

Reviewer #1: I appreciate the scope of the manuscript and the underlying idea but I also wish the authors had made a number of different decisions.

1. I can't understand why the authors use the Abele and Wacjcizke approach to measuring people perceptions but then insist on talking about trust. In this regard, though I don't fully understand why, Abele and his colleagues seem to avoid using the word trust to talk about their research but they seem to only use the word 'trust' in passing (and seem to equate 'trustworthy' with integrity, which almost certainly creates measurement confusion).

This is a good point. We used the four facets structure (i.e., the one of the DPM model) because the most prominent researchers in this area agreed (https://doi.org/10.1037/rev0000262) that this is the best conceptualization of social evaluations’ structure. That is, each of the Big Two (agency/communion) is further divided into two facets (competence/assertiveness and morality/warmth). Specifically, in the area of research on scientists, it is important to use the four-factor approach as it offers more nuance and a better understanding of which of the core social evaluations relate to trust. To make this clear, we largely expanded our introduction subheading (Social Evaluations as Antecedents of Trust) on pages 4-6.

Please see our response to point 2 regarding the comment on trust.

And, in using Abele et al, I also wish you'd reported a CFA to show that the two facets for each of the big two are, indeed, distinct (noting that Abele does not report a CFA in the article cited, either). I have a lot of doubts about the degree to which the two facets are consistently distinct, though it's interesting that one facet in each dimension is about how someone IS (i.e., moral, or able) whereas the other dimension seems to be about how they SEEM (i.e., warm, assertive).

For the current purposes, I'd prefer to see a CFA in an appendix or a full measurement model as the primary results. I think it might also be worth reporting a version of the results that just uses the big two. You might be missing something by measuring competence and assertiveness separately, for eample.

This is a good point as well. As per the response above – it is important to note that the four-facets model of social evaluations we used is theoretically grounded, agreed upon by many of the involved researchers (who also had different ideas of how the structure might look like; https://www.pnas.org/doi/abs/10.1073/pnas.1906720117). In this sense, we simply employed this model to investigate how these dimensions relate to trust, rather than testing whether this structure also applies to the perceptions of scientists. Nevertheless, we conducted additional analyses to check for the factor structure.

We tested two models: one with the four-factor (the one we currently use) and one with the two-factor (Big 2) solution. Only the four-factor solution showed good fit (CFI = 0.946, TLI = .937, RMSEA = .077, SRMR = .052, AIC = 444328, BIC = 444811), but not the two-factor one (CFI = 0.748, TLI = .717, RMSEA = .163, SRMR = .140, AIC = 483346, BIC = 483792). We also compared the models, and the four-factor solution was significantly better (χ2diff(5) = 9592.5, p < .001). We report this in the footnote on p. 10.

We also conducted analyses with the Big Two only, and now report it in the manuscript (p.19) and the supplement (p.39-40). We added the following in the main text: “We ran all the analyses above (non-preregistered) using only the big two factors – agency (calculated as an average of competence and assertiveness) and communion (calculated as an average of warmth and morality). This showed that both big factors contributed to predicting trust which mediated the effect on influence granting (supplement Figures S2 and S3). Importantly, comparing the results of analyses with two vs. four factors allowed us to discern which parts of the agency and communion contribute to trust (competence and morality) and which do not (assertiveness and warmth).”

2. I can't understand why you measured your criterion variable of 'trust' with such a weak measure. In this regard, you might note that Fiske's SCM has a nice set of outcomes that could have been used. Even better, however, would have been to rely on more-standard-than-DPM Integrative Model of Organizational Trust (which you cite) which makes a really nice distinction between trustworthiness perceptions (i.e., appraisals or beliefs) and trust BEHAVIORS as making oneself vulnerable. From this perspective, the trustworthiness perceptions exist in someone's head whereas behavioral trust exists in the world. In contrast, just asking someone 'how much do you trust someone' isn't clearly a perception or a behavior; it's just too vague a question and is thus going to capture a whole bunch of variance while providing little real insight.

With regard to to your IGT, you seem to be trying to recreate the wheel a bit, with all your work to make a criterion variable that you could have adapted from existing research behavioral trust discussions and avoided the use of a single-item measure.

Ultimately, my suggestion is to redo everything and ditch the "how much do you trust" question entirely. It just adds noise. You should also recognize that a limitation of your work is that you should have put more attention into measuring the criterion variable.

This is a good and complex point. Please first note that behaviors/outcomes in the SCM do not relate to trust (e.g., facilitative, harmful behaviors), so we could not have selected those.

We acknowledge the point about the one-item question on trust, however, we believe this is an appropriate measure of trust. First, a lot of authoritative and large-scale research on trust in scientists indeed used one-item measures. Specifically, we took the phrasing from Algan et al., 2021 who investigated trust in scientists in 12 countries with 54 000 participants (https://www.pnas.org/doi/abs/10.1073/pnas.2108576118). Even larger research used the same approach, which is relatively common in this field. For example, Welcome Global Monitor framed the question as follows: How much do you trust [scientists in your country]? 

The reason for such a simple, yet effective question is twofold. Firstly, it does not impose any definitions on what trust is, but the participant is allowed to decide what the term means to themselves. Secondly, it is a relatively cost-efficient way to measure trust, which was important for us given the large number of participants sampled (more than 3000 participants). To make sure that one-item measure of trust is valid, we included several measures of trust in the pilot studies. Across three pilot studies, for trust in different scientific occupations, we found that the one-item measure and the four-item measure of trust (trusting or distrusting scientists to “create knowledge that is unbiased and accurate”, “create knowledge that is useful”, ”, “advise government officials on policy?”, and “inform the public on important issues?”.) have high correlations, mostly around r between 0.6 and 0.7. This is given in Table S2 in the supplement (correlations between 2. and 3. measure)

Importantly, we indeed wanted to make a chain model: social evaluations -> perceptions of trust -> trusting behavior. Because there is no one-to-one translation from social evaluations to trust (i.e., appraisals or beliefs), which we see by beta coefficient sizes, we consider it important to include both perception of trust and behaviors (IGT) in the model, rather than leaving out the trust measure. We also did not want to use Meyer’s model as these factors are highly intercorrelated and are less clearly conceptualized than social evaluation models (they also focus on organizations, rather than individuals/groups).

In the revised version, we included the information above in the introduction (p. 4-6), method (p. 8-9) discussion (p.21), and limitations sections (p. 22).

3. With regard to the criterion variable itself, I wish you had made it a little more specific. You say you tried to write it so that it fit every different type of scientist but, in doing so, you also made it a bit weird in that it's not clear from what's written whether the scientist is providing advice on a topic that's relevant to their area of expertise. The trust/trustworthiness literature in this regard makes the point that the expertise, for example, needs to match the behavior. The fact that someone knows a lot about meteorology isn't especially relevant to whether I would trust them to provide advice on vaccine saftey. The problem you describe, in this regard, seems to something like climate change (vs. a vaccine, or banking regulations).

This is a good point. We wanted to make the task consistent across occupations so that we could test the same model for different occupations and see whether the effects on IGT vary across the occupations. Additionally, it would be very demanding to develop and pilot test 45 scenarios for IGT, while it would not be possible to control for confounds. We expanded on this rationale in both Method (p. 9) and Limitation sections (p. 22).

Also with regard to the criterion variable, I would really like to have seen your initial work look at other potential correlates that specifically relate to behaviors of trust. With regards to what's reported, it'd be nice to show how other groups generally scored, though I'm not sure if there's a parsimonious way to do that.

In the revision, we attempted to report how other groups scored in a parsimonious way by giving the overall means (across participants and occupations) for each group in the Method section (p.9): “On average (across all scientific occupations), participants distributed the points in the following descending order: scientists (25.5), citizens (18.1), community leaders (16.9), themselves (16.6), friends (8.0), politicians (7.5), family (7.3).”

More minor things

- Better label the figures (e.g., the X axis could be Parameter Estimate of Competence on IGT). If you don't read the note first then it'd be easy to think that the figures are means. And, it'd be really nice to have the means in the main body. Given PLOS is online, I'd also suggest making each of the figures bigger.

Thank you for these suggestions, we now implemented all of them. The X-axes for figures are now labeled Parameter Estimate of [Social Evaluation] on Trust/IGT; the means are now moved to the main body. We also made the figures bigger by separating them from one figure (with 4 plots) into two figures (with 2 plots).

- Maybe tone down the use of the word 'effects' and use things like relationships or correlation or estimates to recognize that you're not doing an experiment

This is a good suggestion, we added the terms “prediction/predicting” and “estimates” to tone down the language.

-I do not understand table s2

We now changed the description of the table to clarify it.

- table s3 could have SEs

This is a good suggestion, we now added SEs (although note that SEs are almost the same across occupations). Also, note that this table is now Table 1 in the main text.

-Give all your measures, not just examples; provide a correlation matrix of all of 16 in the supplementary files, if they're really distinct.

We now added all the measures and their correlation matrix (heatmap) of all 20 items (note there are 20, not 16). The correlations support our conclusion to use four facets.

Reviewer #2: The manuscript addresses a relevant question, namely, the psychological determinants of trust in scientists. The (pre-registered) approach is transparent as well as comprehensible. The measurement of trust (on one item) could have been more differentiated, e.g. by measuring trust determinants as introduced by Mayer et al. 1995. For me, it remains unanswered why social evaluation has such a different effect on self-reported trust vs. behavior-based trust (IGT). This could have been discussed in more detail in the discussion. Overall, a highly relevant as well as methodologically clear contribution. I recommend to accept the submission.

We thank you for this review of the paper. In the revision, we try to discuss more the different effects regarding self-reported trust and IGT (p. 21) and also discuss more the limitations of these measures (p. 22; please also see the response to the second point of reviewer #1). We also made changes in the introduction/method to explain our use of social evaluation models (please see the response to the first point of reviewer #1).

Reviewer #3: Thanks for the opportunity to read firsthand about this research. Overall, the research topic is interesting. I have the following suggestions for the author’s reference:

1. The study design should be clearer, why is each person assessed against four randomly selected occupations? And where should each direct impression subject come from? It is suggested that the author should clarify the content.

2. After all, the subjects who answer the questions are very different. It is recommended that the authors conduct a differential analysis on the attributes of the people who answer the questions. In addition, it is recommended that the narrative description of the attributes of the respondent be presented in a table, which will be more clear.

3. The results should be discussed more fully.

In the revision, we added more information in the results section and discussion section. Please see our response to previous reviewers.

4. The review of the literature should make a stronger case for why it is appropriate to select these traits for scientists. It must also be explained why it matters whether scientists trust non-scientists.

We now expanded the introduction (p. 4-6) more to explain why we used the DPM model. Please see the response to reviewer’s #1 point 1.

---

## [Decision Letter · Decision Letter 1]

18 Jan 2024

PONE-D-23-21811R1How Social Evaluations Shape Trust in 45 Types of ScientistsPLOS ONE

Dear Dr. Gligorić,

Thank you for submitting your manuscript to PLOS ONE. After careful consideration, we feel that it has merit but does not fully meet PLOS ONE’s publication criteria as it currently stands. Therefore, we invite you to submit a revised version of the manuscript that addresses the points raised during the review process.

We look forward to receiving your revised manuscript.

Kind regards,

Claudia Noemi González Brambila, Ph.D.

Academic Editor

PLOS ONE

Journal Requirements:

Reviewers' comments:

Reviewer's Responses to Questions

**Comments to the Author**

1. If the authors have adequately addressed your comments raised in a previous round of review and you feel that this manuscript is now acceptable for publication, you may indicate that here to bypass the “Comments to the Author” section, enter your conflict of interest statement in the “Confidential to Editor” section, and submit your "Accept" recommendation.

Reviewer #1: All comments have been addressed

2. Is the manuscript technically sound, and do the data support the conclusions?

Reviewer #1: Yes

3. Has the statistical analysis been performed appropriately and rigorously? 

Reviewer #1: Yes

4. Have the authors made all data underlying the findings in their manuscript fully available?

Reviewer #1: Yes

5. Is the manuscript presented in an intelligible fashion and written in standard English?

Reviewer #1: Yes

6. Review Comments to the Author

Reviewer #1: I guess it's good; It's really interesting data and is well presented. I just still wish you didn't use a direct measure of trust OR called out the fact it's probably a problematic and then compared it to your better measure.

1. The argument that a direct, single-measure trust measures is validated isn't really clear.

2. The argument that it's valid because it's used a lot by others (including non-academic researchers) isn't very compelling.

3. Your argument that's common method variance is creating high correlations is probably partly correct but the more likely problem is that you're comparing at set of specific attitudes to a general attitude rather than an unclear measure that's probably capturing a bit of behavior and a bit of attitude. It's like you ask, how much do you like rap, jazz, R&B, rock, etc. and then correlated it with 'how much do you like music' (vs. would you be willing to go to a show/buy a song). Of course the attitude-attitude (or belief-belief) is going to be higher than the belief-behavioral willingness.

4. Your argument that Fiske's BIAS model doesn't include a behavior isn't quite right. What are active/passive facilitation and harm if not behaviors?

5. Your comment that Besley et al 2021 doesn't include a measure of trust is incorrect. Table 2 and Table 5 include both a direct measure of trust and a trust-as-vulnerable-behavior measures. As with your study, it looks like there's a higher correlation between the direct measure (i.e., attitude to attitude) and the trustworthiness measures than the vulnerability-trustworthiness measures.

6. I still think you should point out that warmth and assertiveness (perceptions about how people BEHAVE) are qualitatively different than morality and competence (perceptions about how people ARE), unless I missed it.

7. PLOS authors have the option to publish the peer review history of their article (what does this mean?). If published, this will include your full peer review and any attached files.

Reviewer #1: No

---

## [Author Response · Author response to Decision Letter 1]

26 Jan 2024

Respond to reviewers letter is attached as a separate document.

---

## [Decision Letter · Decision Letter 2]

15 Feb 2024

How Social Evaluations Shape Trust in 45 Types of Scientists

PONE-D-23-21811R2

Dear Dr. Gligorić,

We’re pleased to inform you that your manuscript has been judged scientifically suitable for publication and will be formally accepted for publication once it meets all outstanding technical requirements.

Kind regards,

Claudia Noemi González Brambila, Ph.D.

Academic Editor

PLOS ONE

Additional Editor Comments (optional):

Reviewers' comments:

Reviewer's Responses to Questions

**Comments to the Author**

1. If the authors have adequately addressed your comments raised in a previous round of review and you feel that this manuscript is now acceptable for publication, you may indicate that here to bypass the “Comments to the Author” section, enter your conflict of interest statement in the “Confidential to Editor” section, and submit your "Accept" recommendation.

Reviewer #1: All comments have been addressed

2. Is the manuscript technically sound, and do the data support the conclusions?

Reviewer #1: Yes

3. Has the statistical analysis been performed appropriately and rigorously? 

Reviewer #1: Yes

4. Have the authors made all data underlying the findings in their manuscript fully available?

Reviewer #1: Yes

5. Is the manuscript presented in an intelligible fashion and written in standard English?

Reviewer #1: Yes

6. Review Comments to the Author

Reviewer #1: (No Response)

7. PLOS authors have the option to publish the peer review history of their article (what does this mean?). If published, this will include your full peer review and any attached files.

Reviewer #1: No

---

## [Editor Report · Acceptance letter]

5 Apr 2024

PONE-D-23-21811R2 

PLOS ONE

Dear Dr. Gligorić, 

I'm pleased to inform you that your manuscript has been deemed suitable for publication in PLOS ONE. Congratulations! Your manuscript is now being handed over to our production team.

Kind regards, 

on behalf of

Dr. Claudia Noemi González Brambila 

Academic Editor

PLOS ONE